# The Acute Effects of Attaching Chains to the Barbell on Kinematics and Muscle Activation in Bench Press in Resistance-Trained Men

**DOI:** 10.3390/jfmk7020039

**Published:** 2022-05-04

**Authors:** Roland van den Tillaar, Atle Hole Saeterbakken, Vidar Andersen

**Affiliations:** 1Department of Sports Science, Nord University, 7601 Levanger, Norway; 2Faculty of Education, Arts and Sports, Western Norway University of Applied Sciences, 6851 Sogndal, Norway; atle.saeterbakken@hvl.no (A.H.S.); vidar.andersen@hvl.no (V.A.)

**Keywords:** variable resistance, barbell velocity, sticking region, EMG, pectoralis major

## Abstract

The aim of the study was to investigate the acute effects of attaching chains on barbell kinematics and muscle activation in the bench press. Twelve resistance-trained men (height: 1.79 ± 0.05 m, weight: 84.3 ± 13.5 kg, one repetition maximum (1-RM) bench press of 105 ± 17.1 kg) lifted three repetitions of bench press in three conditions: (1) conventional bench press at 85% of 1-RM and bench press with chains that were (2) top-matched and (3) bottom-matched with the resistance from the conventional resistance lift. Barbell kinematics and the muscle activity of eight muscles were measured at different heights during lowering and lifting in the three conditions of the bench press. The main findings were that barbell kinematics were altered using the chains, especially the 85% bottom-matched condition that resulted in lower peak velocities and longer lifting times compared with the conventional 85% condition (*p* ≤ 0.043). However, muscle activity was mainly only affected during the lowering phase. Based upon the findings, it was concluded that using chains during the bench press alters barbell kinematics, especially when the resistance is matched in the bottom position. Furthermore, muscle activation was only altered during the lowering phase when adding chains to the barbell.

## 1. Introduction

Traditionally, resistance training is performed with equipment with a constant load (e.g., free weights). When conducting resistance training using free weights, the determination of success or failure is often limited to a small region of the movement referred to as the sticking region [1,2,3,4]. Therefore, different training strategies and instruments to overcome the sticking region have been suggested [5].

Variable resistance has been introduced as an alternative to constant resistance and a strategy to overcome the sticking region. Variable resistance can be defined as a modality where the resistance/load varies throughout the range of motion [6,7]. It has been reported that, when conducting resistance training at maximal intended velocities, variable resistance leads to faster acceleration and a shorter deceleration phase of the barbell [8]. Furthermore, when analyzing the different parts of the repetition, differences in kinematics between variable and constant resistances have been reported. For example, Saeterbakken et al. [9] reported that variable resistance training led to greater velocities in the pre-sticking region, while constant resistance showed greater velocities in the post-sticking region. Such phase-specific effects have also been reported for muscle activation. For example, both Aboodarda et al. [10] and Israetel et al. [11] reported that variable resistance increased the activation of the agonists compared to constant resistance. However, the difference was only observed in the parts of the movement where the variable resistance was greatest.

Variable resistance can be induced by different equipment, such as elastic bands, chains, and specialized machines [12]. Of these, chains are the easiest to implement in training, by just attaching the chains to the barbell. Several studies have examined the acute effects of attaching chains to the barbell; however, most of these were studies in exercises for the lower body [13,14,15,16,17,18]. To the best of our knowledge, only two studies have examined upper body exercises [19,20]. Baker and Newton [19] investigated the effect of chains on barbell kinematics and found that the use of chains with an equivalent of about 15% of 1-RM attached to a barbell at 60% of 1-RM allowed the athletes to generate 10% greater velocities as compared to a conventional barbell load at 75% of 1-RM during the bench press. Godwin et al. [20] compared bench press throws at 45% of 1-RM using either a constant load or a constant load (30% of 1-RM) + chains (15% of 1-RM) in rugby players. The findings showed greater barbell acceleration and velocity for the chain condition. In both of these abovementioned studies, the same absolute load (in kg) was lifted, with the load being matched in the top position (extended arms). Consequently, in the bottom position (barbell at sternum), the resistance in the chain condition would be less than in the free weights condition, which could explain the findings of the studies. Furthermore, none of these studies included electromyography (EMG), which could provide more information about the potential mechanisms involved with chains. To date, no study has compared the acute effects when matching the resistance in the bottom position of the movement with EMG measurements. It could be rationalized that this would affect both kinematics and muscle activity and therefore be of great scientific and practical interest.

Therefore, the aim of the study was to investigate the acute effects of (i) attaching chains to the barbell on kinematics and muscle activation in the bench press and (ii) investigating differences between matching the resistance in the top or the bottom position. It was hypothesized that the use of chains would alter the barbell kinematics and muscle activation compared to the conventional bench press due to the increased and decreased loads during the lowering and lifting of the barbell. More specifically, matching the resistance in the top position was hypothesized to increase barbell velocity, especially in the first part of the movement, while matching the resistance in the bottom position would increase muscle activation.

## 2. Methods

### 2.1. Experimental Approach to the Problem

To investigate the effect of attaching chains to a barbell upon kinematics and muscle activation in the bench press, a within-subject design was used in which each participant bench pressed in three conditions: conventional bench press at 85% of one repetition maximum (1-RM) and bench press with chains in which the load at either the top or the bottom of the lift was similar to the load of the conventional bench press at 85% of 1-RM. The dependent variables were the barbell kinematics and muscle activation at different heights during the lowering and lifting phases.

#### 2.1.1. Participants

Twelve resistance-trained males (height: 1.79 ± 0.05 m, body mass: 84.3 ± 13.5 kg/m^2^ and 1-RM bench press: 105 ± 17.1 kg), with at least one year of regular (1–2 times per week) bench press training, participated in this study. Written consent was obtained from each participant at the start of the study after informing each participant about the procedures and risks of the experiment. The study was conducted following the latest revision of the Declaration of Helsinki and current ethical regulations for research and was approved by the National Center for Research Data (pr.nr: 991974).

#### 2.1.2. Procedure

Most participants did not have any experience with chains during a bench press. Therefore, a familiarization session was performed one week before the test session. Of note, 1-RM testing was not conducted in this familiarization session since all participants had participated in other bench press studies over the last four weeks examining the 1-RM. In the familiarization session, 2–3 sets consisting of 2–3 reps with chains were performed. The familiarization was conducted with loads of 85% of 1-RM matched at both the top and the bottom of the lift. To calculate the correct top and bottom loads with the chains, the participant had to keep the barbell with chains at these positions. The total force was measured by a force plate (Ergotest Innovation, Stathelle, Norway) placed under the bench on which the participant’s entire body was placed. The chains (Pivot Fitness, Tianjin, China) attached to the barbell were each 12 kg and 0.75 m long. They were placed upon boxes (0.4 m in height (Figure 1)) which resulted in a load of +5 kg and −5 kg compared to the 85% of 1-RM load during the conventional condition. For example, lifting 100 kg in the conventional lift, 95 kg (barbell + chains) would represent the top-matched condition, while 105 kg (barbell + chains) would represent the bottom-matched condition. Thereby, the variable resistance in the chain conditions consisted of an average of 5.1% compared with the conventional load.

At the experimental session, body height and mass were measured. Afterward, the skin was prepared (shaved, washed with alcohol, and abraded) for the placement of gel-coated surface EMG electrodes. The electrodes were placed on the right side of the body. The electrode pads (11 mm contact diameter, 20 mm center-to-center distance) were placed in the presumed direction of the underlying muscle fibers, with a center-to-center distance of 2.0 cm. Self-adhesive electrodes (Dri-Stick Silver circular sEMG Electrodes AE-131, NeuroDyne Medical, Cambridge, MA, USA) were positioned on the bellies of the following eight muscles: pectoralis (clavicular and sternal heads), triceps brachii (medialis, lateralis, and longus), biceps brachii, and deltoid (lateral and anterior). Electrodes were placed according to the recommendations of SENIAM [21]. After the placement of the electrode pads, the participant performed the warm-up. The warm-up was individualized to ensure that the participant was ready to lift in the three conditions. Basically, the warm-up consisted of two sets with light loads and many reps (10–12 reps), then two sets of increasing heavy loads with few reps (6–8 reps), and finally 5 min of rest before starting the test.

The participants were instructed to lower and lift the barbell as quickly as possible but under full control. The barbell had to touch the chest before lifting it to full lockout, without bouncing. The grip placement was individually selected but had to be at least outside shoulder width and inside the markers of maximal allowed grip width in a powerlifting competition. This position was marked and controlled for the entire test session with the different conditions. The participant was instructed to lift one set of three reps in each of the three conditions: (1) 85% of 1-RM conventional lift, (2) chains with matched load at top and (3) at the bottom of lift at 85% of 1-RM conventional lift. The order of the different conditions was stratified randomly for the participants, resulting in six evenly balanced groups of two participants with their own order of lifting conditions. Between each attempt, 4–7 min of rest was prescribed depending on the personal needs of the participants to avoid fatigue.

#### 2.1.3. Measurements

Muscle activation was measured with a surface electromyography (EMG) system (Musclelab v.10.190, Ergotest Innovation, Stathelle, Norway) with a sampling rate of 1000 Hz. To minimize noise induced from external sources, the EMG raw signal was amplified and filtered using a preamplifier located as near the pickup point as possible. The common-mode rejection ratio (CMRR) was 106 dB, and the input impedance between each electrode pair was >10^12^ Ω. Signals were band-pass (fourth-order Butterworth filter) filtered with a cut off frequency of 20 Hz and 500 Hz, rectified, integrated, and converted to root-mean-square (RMS) signals using a hardware circuit network (frequency response 450 kHz, averaging constant 12 ms, total error ± 0.5%) [22]. To locate possible region-specific differences in muscle activity between the three conditions, the lifting height was divided into eight equal parts (percentage of each phase) in the lowering and lifting phases. To compare the EMGs of the different lifting conditions, EMGs were normalized with the highest RMS per height measured during one of the three conditions for each participant. Barbell kinematics were measured with a linear encoder (Ergotest Innovation, Stathelle, Norway) attached to the barbell to measure bar velocity, vertical displacement, and timing. The linear encoder measured barbell vertical displacement and time with a resolution of 0.019 mm and a sampling rate of 200 Hz. Both lowering and lifting phases of the second repetition were analyzed since, in the first rep, participants reserved themselves [22] and the third rep was not always possible due to failure in the bottom-matched chain condition. Peak lowering velocity (v_max down_), first (v_max1_) and second peak velocities (v_max2_), and minimal velocity (v_min_) (sticking region events) were identified, together with their distance and timing (absolute and relative) and total lifting time of both phases to analyze the changes of barbell kinematics during the three conditions. The linear encoder and EMG were synchronized and analyzed in Musclelab v.10.212.97.5176 (Ergotest Innovation, Stathelle, Norway).

#### 2.1.4. Statistical Analyses

To assess changes in barbell kinematics between the three conditions, a one-way analysis of variance (ANOVA) with repeated measures was performed on each of the variables (velocity, distance, and time at the different events). To investigate the effect of the condition on the EMG a 3 (condition) × 8 (height) ANOVA with repeated measures was performed for each phase and each muscle. In cases where the sphericity assumption was violated, the Greenhouse-Geisser adjustments of the *p*-values were reported. When significant differences were found, a Holm-Bonferroni post hoc comparison was performed. The level of significance was set at *p* < 0.05. Statistical analyses were performed in SPSS version 27.0 (IBM Corp. Released 2020. IBM SPSS Statistics for Windows, Armonk, NY, USA). All results are presented as mean ± SD. The effect size was evaluated with eta partial squared (η^2^) where 0.01 < η^2^ < 0.06 constitutes a small effect, a medium effect when 0.06 < η^2^ < 0.14, and a large effect when η^2^ > 0.14 [23].

## 3. Results

The participants lifted 83.2 ± 14.2 kg in the 85% conventional bench press. A significant effect of the condition was found for the total lowering and lifting times (F ≥ 3.8, *p* ≤ 0.039, η^2^ ≥ 0.24), the velocity at each event (F ≥ 3.6, *p* ≤ 0.043, η^2^ ≥ 0.23), distance, relative distance and relative time at v_max1_ (F ≥ 3.7, *p* ≤ 0.039, η^2^ ≥ 0.24), and the interval from v_max1_ to v_min_ and v_min_ to v_max2_ (F ≥ 4.1, *p* ≤ 0.030, η^2^ ≥ 0.25), but not in relative timing, relative distance, and distance at these two events (F ≤ 1.8, *p* ≥ 0.19, η^2^ ≤ 0.13). A post hoc comparison revealed that velocity was significantly lower at each event for the chain 85% bottom-matched condition compared with the conventional 85% condition and in the downward phase, which was also lower than with the chain 85% top-matched condition. In addition, the total lowering time was significantly longer in the chain 85% bottom-matched condition when compared to that in the conventional 85% lifts and the lifting phase with both conditions (Table 1). The interval times v_max1_−v_min_ and v_min_−v_max2_ were significantly longer when lifting with the chains 85% bottom-matched condition when compared with the conventional 85% condition, while the relative time of occurrence of v_max1_ and v_max2_ were, respectively, earlier and later during the lift than with the conventional 85% condition. The distance at which v_max1_ occurred was significantly lower at chains in the bottom-matched condition when compared with that in the other two conditions, which also resulted in a significantly lower relative distance between the chain 85% bottom-matched and the chain 85% top-matched conditions (Table 1).

During the lowering and lifting phases, muscle activity changed significantly for all muscles (F ≥ 4.4, *p* ≤ 0.31, η^2^ ≥ 0.33), except for the sternal head of the pectoralis major in both phases (F ≤ 1.4, *p* ≥ 0.26, η^2^ ≤ 0.11) and the long and medial heads of the triceps brachii during the lowering phase (F ≤ 2.6, *p* ≥ 0.082, η^2^ ≤ 0.19). However, a significant effect of the condition was found during the lowering phase for the medial and long heads of the triceps brachii (F ≥ 3.1, *p* ≤ 0.046, η^2^ ≥ 0.27) and lateral deltoid (F = 6.4, *p* = 0.007, η^2^ = 0.37). Additionally, significant interaction effects were found during the lowering phase for both parts of the deltoid and the pectoralis muscles (F ≥ 2.0, *p* ≤ 0.021, η^2^ ≥ 0.15). For the lifting phase, no significant condition (F ≤ 2.17, *p* ≥ 0.138, η^2^ ≤ 0.16) or interaction effects (F ≤ 1.38, *p* ≥ 0.169, η^2^ ≤ 0.11) were found for any of the other muscles (Figure 2 and Figure 3). Post hoc comparison revealed that the muscle activity of the long head of the triceps brachii and lateral deltoid were significantly higher in the 85% bottom-matched condition when compared with those in the other two conditions, while the EMG activity of the medial head of the triceps brachii in the 85% top-matched condition was significantly lower than that in the conventional 85% condition during the lowering phase (Figure 2). Furthermore, during parts of the lowering phase, both parts of the pectoralis showed that the 85% bottom-matched condition had a higher activation than the other conditions, while the deltoid muscles also showed a different development of activity between the three conditions (Figure 3).

## 4. Discussion

The aim of the study was to investigate the acute effects on kinematics and muscle activation when attaching chains to a barbell while performing the bench press. The main findings were that barbell kinematics were altered using the chains, especially in the 85% bottom-matched condition, resulting in lower peak velocities and longer lifting times compared with the conventional 85% condition. However, EMG activity was mainly only affected during the lowering phase for all muscles, except the biceps brachii.

The 85% bottom-matched condition resulted in lower velocities in all events, which is most likely explained by the increased resistance. When the arms were straight in the bottom-matched condition, the total load was 90.1% of 1-RM. Consequently, this requires more force output from the muscles to lift the load while lowering and lifting it again. The present findings were supported by van den Tillaar and Kwan [24] and Kristiansen, et al. [25] who showed that extra loads during the lowering phase of the bench press resulted in lower peak velocities. Furthermore, the vertical displacement of the sticking region in the present study also starts a bit earlier in the 85% bottom-matched condition when compared to those in the other conditions (0.04 m vs. 0.05 m, Table 1) due to the lower peak velocity in the 85% bottom-matched condition. However, the sticking region ends at the same height between the conditions, indicating that the sticking region occurs in a mechanically poor position region [4,26,27].

Surprisingly, no alterations were found between the conventional 85% and the 85% top-matched conditions. This finding contradicts Baker and Newton [19] who reported increased peak and average velocities for the chain condition. This discrepancy with Baker and Newton [19] is probably due to the loads that were examined. In the present study, 85% of 1-RM had to be lifted, while the athletes in the study of Baker and Newton [19] lifted 75% of 1-RM [19]. Due to the lower load in the study of Baker and Newton [19], the participants could increase the peak velocity at the start of the lifting phase during the chain condition without a sticking region occurring. However, in the present study, the load was already so high that, in the 85% top-matched condition, a sticking region also occurs, resulting in a performance similar to that in the conventional 85% lift. In addition, the difference in load from the bottom- to top-matched positions with chains was much higher in the study of Baker and Newton [19] than in the present study (15% vs. 5.1%), which also could influence the lifting velocity. This speculation is supported by previous studies [13,14]. For example, Heelas et al. [14] compared lifting velocity in the deadlift between using only free weights and using free weights combined with a different percentage of variable resistance. The authors reported that the mean velocity increased as the contribution from the variable resistance was increased. Contrary to this, Berning et al. [13] found no difference in velocity when comparing the Olympic clean and jerk using either free weights or free weights + chains. Importantly, and similar to the present study, the contribution from the chains in Berning et al. [13] was approximately 5%, which the authors suggest may be too little to elicit an effect.

The alteration in the barbell kinematics between the conditions was not accompanied by similar changes in muscle activation (Figure 2 and Figure 3). Only during the lowering phase did the 85% bottom-matched condition result in higher activation of the lateral deltoid and long head of the triceps compared to the 85% top-matched condition. However, the differences were not apparent at the start of the lowering phase but were apparent in the whole or other parts of the lowering phase. Furthermore, only the anterior deltoid and the sternal part of the pectoralis major showed a pattern of increased activity at the start of the lowering phase during the 85% bottom-matched condition, which disappears when lowering the barbell compared to the conventional 85% condition. The medial head of the triceps showed decreased activity when lifting with the 85% top-matched condition compared with the conventional 85% condition. The lack of differences in muscle activation between the conventional 85% and the 85% top-matched conditions during the lowering phase may be due to increased stability requirements from the chains. Since the chain parts reach the boxes, this could cause some disturbances and jerks in the horizontal direction. This may cause more muscle activation in the 85% top-matched condition and thereby lower the expected lifting performance. Importantly, in general, the levels of activation were low during the lowering phases. Therefore, the importance of these findings is questionable.

The fact that no significant difference in any of the muscles was found during the lifting phase (Figure 2 and Figure 3) was surprising since the barbell kinematics (Table 1) changed. An explanation could be that, during the lifting phase, muscles were maximally activated, especially when the lifts reached full exhaustion [28]. Furthermore, the differences in loads between the three conditions were low (maximal 10% between the 85% top- and bottom-matched conditions). With this small difference in loads, muscles are activated at the same level [29]. When lifting with different loads with maximal intention in the bench press, van den Tillaar and Souza [30] found that muscle activity did not change between loads of 10% difference. Of note, previous studies have argued that, when aiming to optimize the neuromuscular stress, a relatively large contribution should come from the variable component [10,31].

The present study has some limitations. Firstly, none of the participants had any experience performing the bench press with chains. Although they had one familiarization session, it may have not been enough to diminish the possible learning effects. Consequently, the lifts with chains may not have been performed maximally as is probable with the 85% top-matched condition. Secondly, there is always the possibility of cross-talk when measuring EMG, which could influence the EMG results [32]. In future studies, more familiarization sessions or a training period should be included to investigate if the absence of differences in barbell kinematics between the 85% top-matched and the conventional 85% condition was the result of extra stability requirements or the effect of a lack of experience.

## 5. Conclusions

Based on the findings of the present study, it was concluded that using chains during the performance of the bench press alters barbell kinematics, especially when the resistance is matched in the bottom position. Furthermore, muscle activation was only altered during the lowering phase when adding chains to the barbell.

### Practical Application

Variable resistance has been suggested to alter the kinematics and neuromuscular stress when compared to constant resistance. The result of the present study shows that attaching chains to the barbell in the bench press changes the kinematics compared to using only free weights. Importantly, these differences were mainly limited to the chain condition where the resistance was matched in the bottom position of the movement. The differences in muscle activation were less pronounced and limited to mainly the lowering phase. Therefore, athletes and recreational actives that want to focus on lower barbell velocities and longer lifting times should add chains to the barbell and match the resistance with their regular bench press resistance in the bottom position of the movement. However, based on the findings from previous studies, variable resistance could also be an alternative for increasing barbell velocity [14,19] and neuromuscular stress [10,31]. Importantly, it appears that, to obtain this, the percentage contribution from variable resistance should be higher than that in the present study.

## Figures and Tables

**Figure 1 jfmk-07-00039-f001:**
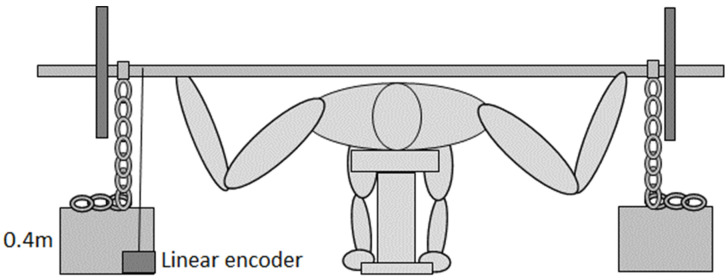
Bench press set up with chains and linear encoder attached to the barbell.

**Figure 2 jfmk-07-00039-f002:**
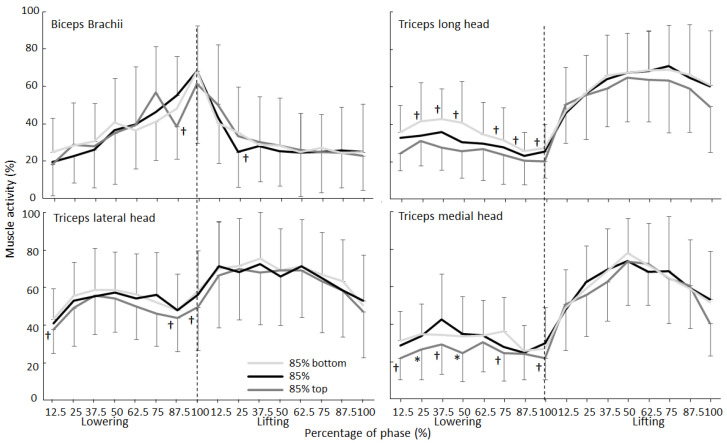
Mean (±SD) RMS EMG per 12.5% of the total barbell distance in the lowering and lifting phases for the biceps and triceps brachii muscles during the bench press per condition. * Indicates a significant difference with the other two conditions at this height on a *p* < 0.05 level. † Indicates a significant difference with the condition furthest from this condition away at this height on a *p* < 0.05 level.

**Figure 3 jfmk-07-00039-f003:**
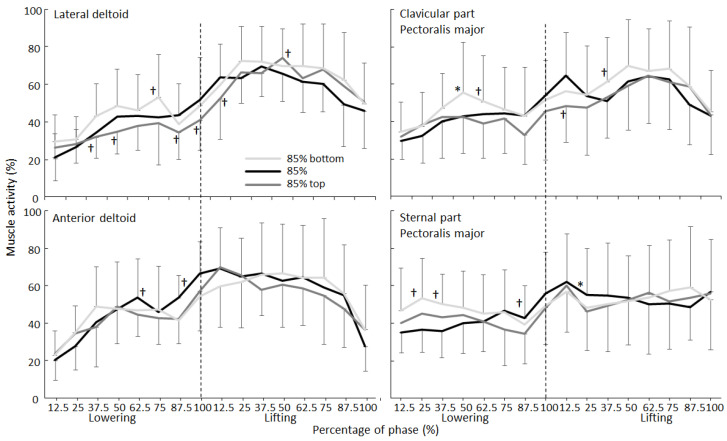
Mean (±SD) RMS EMG per 12.5% of the total barbell distance in the lowering and lifting phases for the deltoid and pectoralis major muscles during the bench press per condition. * Indicates a significant difference with the other two conditions at this height on a *p* < 0.05 level. † Indicates a significant difference with the condition furthest from this condition away at this height on a *p* < 0.05 level.

**Table 1 jfmk-07-00039-t001:** Barbell kinematics of the three conditions at second repetition (mean ± SD).

Parameter	Conventional 85%	Chains 85% Top	Chains 85% Bottom
v_max down_ (m/s)	0.60 ± 0.15	0.57 ± 0.13	0.51 ± 0.10 *
Time v_max down_ (s)	0.49 ± 0.24	0.52 ± 0.25	0.51 ± 0.18
Relative time v_down_ (%)	48.5 ± 18.1	46.6 ± 13.7	43.6 ± 9.5
Total lowering time (s)	1.00 ± 0.21 †	1.09 ± 0.32	1.15 ± 0.28 †
v_max1_ (m/s)	0.37 ± 0.10 †	0.37 ± 0.07	0.32 ± 0.10 †
Distance v_max1_ (m)	0.05 ± 0.02	0.05 ± 0.03	0.04 ± 0.02 *
Time v_max1_ (m)	0.18 ± 0.06	0.19 ± 0.06	0.16 ± 0.05
Relative distance v_max1_ (%)	13.7 ± 5.6	14.9 ± 7.0 †	10.7 ± 4.7 †
Relative time v_max1_ (%)	15.3 ± 6.5	14.9 ± 7.6	9.2 ± 6.0 *
v_min_ (m/s)	0.26 ± 0.14 †	0.22 ± 0.10	0.13 ± 0.14 †
Distance v_max1_−v_min_ (m)	0.11 ± 0.04	0.12 ± 0.05	0.13 ± 0.05
Interval v_max1_−v_min_ (s)	0.38 ± 0.18 †	0.45 ± 0.21	0.95 ± 0.88 †
Relative distance v_min_ (%)	43.1 ± 10.2	48.8 ± 19.7	47.8 ± 12.4
Relative time v_min_ (%)	44.3 ± 9.0	46.7 ± 17.4	46.8 ± 12.1
v_max2_ (m/s)	0.43 ± 0.13 †	0.38 ± 0.11	0.31 ± 0.12 †
Distance v_min−_v_max2_ (m)	0.15 ± 0.04	0.14 ± 0.07	0.14 ± 0.04
Interval v_min−_v_max2_ (s)	0.50 ± 0.25 †	0.56 ± 0.33	0.89 ± 0.50 †
Relative distance v_max2_	85.0 ± 4.8	87.7 ± 4.6	87.3 ± 5.9
Relative time v_max2_ (%)	82.4 ± 6.6 †	85.1 ± 6.1	86.5 ± 6.7 †
Total lifting time	1.28 ± 0.31	1.42 ± 0.42	2.27 ± 1.26 *
Total distance	0.36 ± 0.04	0.36 ± 0.03	0.35 ± 0.04

† Significant difference between these two conditions on a *p* < 0.05 level. * Significant difference with the other conditions on a *p* < 0.05 level.

## Data Availability

The raw data supporting the conclusions of this article will be made available by the authors, without undue reservation.

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
