# Peer review of "The Acute Effects of Attaching Chains to the Barbell on Kinematics and Muscle Activation in Bench Press in Resistance-Trained Men"

_jfmk, 2022, doi:10.3390/jfmk7020039_

Round 1

Reviewer 1 Report

The study was generally will written with clear justification of the purpose and discussion of results. I only have some minor issues that i hope the authors will be able to clarify and/or amend.

Abstract: The instructions for authors stated that the abstract should be 200 words. The authors will have to reduce the word count. Also, please include statistical values for any significant differences observed.

Line 106-112: What was the reason for having the chains to contribute ~5.1% of the lifting load? And studies that was taken reference to?

Line 127-130: What was the intensity of the load? Was it based on %1RM or repetition in reserve etc?

Line 248-251: This is like a repeat from the previous paragraph.  The authors may consider removing it.

Author Response

Abstract: The instructions for authors stated that the abstract should be 200 words. The authors will have to reduce the word count. Also, please include statistical values for any significant differences observed.

We have reduced the word count and included the p values for the significant differences. More space was not left.

Line 106-112: What was the reason for having the chains to contribute ~5.1% of the lifting load? And studies that was taken reference to?

These were the only chains we could buy. Thereby it resulted in this difference of 5.1 %. It is perhaps not a good reason, but these were the only ones that were commercially available.

Line 127-130: What was the intensity of the load? Was it based on %1RM or repetition in reserve etc?

The exact intensity was not specified for the warmup since it was individualized as all participants were had enough experience to know when they were ready for the test and to avoid that they were not ready. Basically, two sets with low loads (30-50% of 1-RM) were lifted followed by two sets with heavier loads (60-75%). Heavier weights were not used to avoid fatigue. However, some lifted one set less or more when the needed this before the test.

Line 248-251: This is like a repeat from the previous paragraph.  The authors may consider removing it.

We have removed this from the text.

Reviewer 2 Report

Congratulations to the authors for an interesting study which is also highly applicable in the field of training and disciplines where heavy loads such as those studied are of high interest (ie., powerlifting).

Following the limitations commented by the authors, I would suggest to include further information on how the randomisation was performed and how it looked (how many subjects performed each condition first, second and last) in case some differences existed. Otherwise I suggest to indicate how the randomisation was performed and clearly state that every condition was performed as many times as the others first/second/last. This can provide useful information regarding the potential learning-effect mentioned.

Author Response

Following the limitations commented by the authors, I would suggest to include further information on how the randomisation was performed and how it looked (how many subjects performed each condition first, second and last) in case some differences existed. Otherwise I suggest to indicate how the randomisation was performed and clearly state that every condition was performed as many times as the others first/second/last. This can provide useful information regarding the potential learning-effect mentioned.

It was stratified randomized for the participants, resulting in six evenly balanced groups of two participants with their own order of lifting conditions. This is now mentioned in the methods part.